A novel computational approach to the silencing of Sugarcane Bacilliform Guadeloupe A Virus determines potential host-derived MicroRNAs in sugarcane (Saccharum officinarum L.)

Ashraf Fakiha 1
Ashraf Muhammad Aleem muhammad.aleemashraf@iub.edu.pk 2 3
Hu Xiaowen 4
Zhang Shuzhen zhangshuzhen@itbb.org.cn 1
1 Institute of Tropical Bioscience and Biotechnology, Chinese Academy of Tropical Agricultural Sciences , Haikou , Hainan , China
2 Haikou Experimental Station, Chinese Academy of Tropical Agricultural Sciences , Haikou , Hainan , China
3 Department of Plant Breeding and Genetics, University College of Agriculture and Environmental Sciences, Islamia University of Bahawalpur, Baghdad-Ul-Jadeed Campus , Bahwalpur , Pakistan
4 Zhanjiang Experimental Station, Chinese Academy of Tropical Agricultural Sciences , Zhanjiang , Guandong , China
Orlov Yuriy
Electronic publication date: 2020 Jan 13
Publication date: 2020
Volume: 8
Electronic Location ID: e8359
Received 2019 Jul 4; Accepted 2019 Dec 5
Copyright: ©2020 Ashraf et al.
Copyright year: 2020
Copyright holder: Ashraf et al.
License: This is an open access article distributed under the terms of the Creative Commons Attribution License, which permits unrestricted use, distribution, reproduction and adaptation in any medium and for any purpose provided that it is properly attributed. For attribution, the original author(s), title, publication source (PeerJ) and either DOI or URL of the article must be cited.
License URL: https://creativecommons.org/licenses/by/4.0/

Keywords: Computational algorithms, R language, miRNA, Saccharum officinarum, Sugarcane Bacilliform Guadeloupe A Virus, Target prediction, Gene silencing, miRanda, Virus-host interaction, RNA interference

Funding: National Natural Science Foundation of China 31771865 Sugar Crop Research System CARS-170301 ‘Talented Young Scientist Program” (TYSP-4th Batch 2017-2018) of China Project ID Pakistan-18-004 This study is supported by the National Natural Science Foundation of China (No. 31771865), the Sugar Crop Research System (CARS-170301) and the “Talented Young Scientist Program” (TYSP-4th Batch 2017-2018) of China Project ID (Pakistan-18-004). The funders had no role in study design, data collection and analysis, decision to publish, or preparation of the manuscript.

==============================
Sugarcane Bacilliform Guadeloupe A Virus (SCBGAV, genus Badnavirus, family Caulimoviridae) is an emerging, deleterious pathogen of sugarcane which presents a substantial barrier to producing high sugarcane earnings. Sugarcane bacilliform viruses (SCBVs) are one of the main species that infect sugarcane. During the last 30 years, significant genetic changes in SCBV strains have been observed with a high risk of disease incidence associated with crop damage. SCBV infection may lead to significant losses in biomass production in susceptible sugarcane cultivars. The circular, double-stranded (ds) DNA genome of SCBGAV (7.4 Kb) is composed of three open reading frames (ORFs) on the positive strand that replicate by a reverse transcriptase. SCBGAV can infect sugarcane in a semipersistent manner via the insect vectors sugarcane mealybug species. In the current study, we used miRNA target prediction algorithms to identify and comprehensively analyze the genome-wide sugarcane (Saccharum officinarum L.)-encoded microRNA (miRNA) targets against the SCBGAV. Mature miRNA target sequences were retrieved from the miRBase (miRNA database) and were further analyzed for hybridization to the SCBGAV genome. Multiple computational approaches—including miRNA-target seed pairing, multiple target positions, minimum free energy, target site accessibility, maximum complementarity, pattern recognition and minimum folding energy for attachments—were considered by all algorithms. Among them, sof-miR396 was identified as the top effective candidate, capable of targeting the vital ORF3 of the SCBGAV genome. miRanda, RNA22 and RNAhybrid algorithms predicted hybridization of sof-miR396 at common locus position 3394. The predicted sugarcane miRNAs against viral mRNA targets possess antiviral activities, leading to translational inhibition by mRNA cleavage. Interaction network of sugarcane-encoded miRNAs with SCBGAV genes, created using Circos, allow analyze new targets. The finding of the present study acts as a first step towards the creation of SCBGAV-resistant sugarcane through the expression of the identified miRNAs.

Introduction

Sugarcane bacilliform virus (SCBV, genus: Badnavirus, family Caulimoviridae) is a circular, non-enveloped bacilliform, monopartite DNA virus that harbors about 7.5 kb double-stranded (ds) DNA with three open reading frames (ORFs) (Sun et al., 2016). SCBV was first reported in the sugarcane cultivar (B34104) in the South American continent (Autrey et al., 1992) and disseminated into many sugarcane (Saccharum officinarum L.)-growing regions around the globe (Autrey et al., 1992). SCBV is implicated in yield losses in sugarcane cultivars. SCBV infection may lead to significant losses in biomass production in susceptible sugarcane cultivars (Lockhart & Autrey, 2000). SCBV infect sugarcane cultivars utilized for sugar and bioenergy production and constitute a major threat to the global exchange of sugarcane germplasm (Borah et al., 2013). Variable follicle symptoms were observed in some sugarcane varieties showing severe chlorotic mottling, stunted growth and pronounced freckle (Lockhart, Ireyt & Comstock, 1996; Viswanathan, Alexander & Garg, 1996). SCBV has wide host range. It comprises members of plant families Poaceae (sugarcane, johnsongrass, rice, sorghum, Panicum maximum, Rottboellia exaltata and Brachiaraia sp.) and Musaceae (bananas) (Borah et al., 2013; Lockhart & Autrey, 2000; Lockhart, Ireyt & Comstock, 1996). SCBVs are highly endemic in S. officinarum germplasm. These viruses can also infect commercial sugarcane (Saccharum interspecific hybrids), S. barberi, S. robustum, S. spontaneum and S. sinensis (Lockhart, Ireyt & Comstock, 1996). SCBV is transmitted naturally by sap-sucking insect vector mealybug species (Saccharicoccus sacchari and Dysmicoccus boninsis) through vegetative cuttings. Agrobacterium-mediated inoculation is used to disseminate SCBV infection experimentally (Bouhida, Lockhart & Olszewski, 1993; Lockhart, Ireyt & Comstock, 1996; Viswanathan, Alexander & Garg, 1996). Previous studies have revealed that SCBV is a heterogeneous badnavirus species. SCBV exist in nature as a complex of viruses or distinct strains and genetic variants. Currently, four distinct SCBV species have been classified by the International Committee on Taxonomy of Viruses (ICTV). These are Sugarcane Bacilliform Guadeloupe A Virus (SCBGAV), Sugarcane Bacilliform Guadeloupe D Virus (SCBGDV) Sugarcane bacilliform MO virus (SCBMOV-MOR) and Sugarcane bacilliform IM virus (SCBIMV-QLD) originating from Guadeloupe, Morocco and Australia, respectively (Adams & Carstens, 2012; Adams et al., 2016; Bouhida, Lockhart & Olszewski, 1993; Geijskes et al., 2002; Muller et al., 2011).

SCBGAV was first identified as a sugarcane R570 hybrid variety in Guadeloupe in 2011 (Muller et al., 2011). SCBGAV was classified as a new species in the genus badnavirus under the SCBV-A group as assigned by ICTV (Adams et al., 2016). The disease symptoms appear in the form of chlorosis, leaf freckle and mottling. Some infected plants were also observed to exhibit no symptoms. Recently in China, SCBV-infection resulted in reduced sucrose content, juice, stalk weigh, purity and gravity in sugarcane plants (Ahmad et al., 2019).

SCBGAV is a mealybug-transmitted monopartite badnavirus that infects sugarcane and is composed of a 7,444 bp circular, dsDNA genome containing three ORFs on the positive strand that replicate by a virus-encoded reverse transcriptase (RT). The two small proteins of sizes (176–185 and 122–135 amino acids), were encoded by the ORF1 and ORF2 respectively. The ORF3 was a key component of SCBGAV genome, encoded by a polyprotein (1,786–1,933 amino acids) (Muller et al., 2011).

Sugarcane (Saccharum officinarum L.) has inherited an active immunity that is composed of microRNAs (miRNAs) to combat infection. The miRNAs are a class of 21–23 nucleotide-long, endogenous, noncoding RNAs that govern gene regulation and expression at the post-transcriptional level. In plants, this mechanism occurs through the process of mRNA cleavage or translational repression. They are synthesized after the processing of hairpin miRNA precursors using an RNase-III-like enzyme (Dicer) to control gene expression (Brodersen & Voinnet, 2006).

RNA silencing, through miRNAs present in the host plant, thus imparts natural immunity and resistance to the host against foreign genetic elements including plant viruses (Carbonell & Daròs, 2019; Qu, Ye & Fang, 2007).

Viral miRNA degradation is the simplest molecular approach to combat viral infection in plants. Successful demonstration about changes in the nucleotide transcripts of mature miRNAs does not affect their biogenesis and the application of modified miRNAs to target Potyviruses in Arabidopsis was reported first time (Niu et al., 2006). Several studies have been successful in generating resistance against plant viruses using amiRNA-technology in crop plants such as tomato (Chen et al., 2019; Zhang et al., 2011) and cotton (Akmal, Baig & Khan, 2017; Ali et al., 2013). Most recently, amiRNA-mediated gene silencing has been demonstrated against Potexvirus and Tobamovirus successfully (Petchthai, Le Yee & Wong, 2018).

miRNA: mRNA target identification, validation and interactions are the basis for discerning the important functions of miRNAs in the greater perspective of regulatory network leading to different biotic developments. Computational algorithms for identification and analysis of miRNA targets enable the process of narrowing down possible target binding sites for experimental validation at transgenic level. In-silico strategies model how miRNAs target specific mRNAs and a number of algorithms are accessible, each with a diverse approach to miRNA target prediction and identification. Although, it might be valuable to have access to a variety of tools and algorithms with different capabilities, the user is challenged with a significant option in determining which algorithm/tool to apply (Peterson et al., 2014).

The development of artificial microRNAs (amiRNAs)-mediated gene silencing method offers a highly precise, targeted approach that gives several advantages over RNAi-mediated gene silencing strategy. Design, construction and validation of the amiRNAs depend upon the assembly of an endogenous precursor miRNA, which is substituted with a selected miRNA nucleotide transcript complementary to the target sequence (Niu et al., 2006; Ossowski, Schwab & Weigel, 2008).

The objective of the present research work is to predict sugarcane-encoded miRNAs that have the potential to develop resistance against badnaviruses, especially the SCBGAV. The major research objective was designed to identify the potential host-derived miRNAs in the SCBGAV genome and to screen the most promising miRNAs to understand the complex host-virus interactions. The novel amiRNA silencing method has been adopted for the first time to explore as a source of creating resistance to a monopartite badnavirus. The predicted miRNAs may provide help for the generation of SCBGAV-resistant sugarcane plants through genetic engineering in the future.

Materials & Methods

Retrieval of mature miRNAs of sugarcane

The sugarcane (Saccharum-officinarum L) and (Saccharum spp.) plant miRNA sequences were accessed from the microRNA database, miRBase (http://www.mirbase.org/cgi-bin/browse.pl/) (Griffiths-Jones et al., 2006; Griffiths-Jones et al., 2007; Kozomara, Birgaoanu & Griffiths-Jones, 2018; Kozomara & Griffiths-Jones, 2013). A total of 28 miRNAs were retrieved (Table S1).

SCBGAV genome retrieval and annotation

The nucleotide sequence of SCBGAV dsDNA genome (accession number NC_038382.1) composed of 7,444 bp was retrieved from the NCBI database. pDRAW32 plasmid map software (v 1.1.129) was used for annotation of genome features of SCBGAV (Fig. 1).

Figure 1 Genome organization of Sugarcane Bacilliform Guadeloupe A Virus.

Three ORFs are shown with arrows. The double stranded DNA genome of SCBGAV is composed of 7,444 bp.

miRNA target prediction in SCBGAV genome

Four computational algorithms (Table 1)—named miRanda, RNA22, RNAhybrid and psRNATarget—were selected for the screening of mature miRNAs of sugarcane against the SCBGAV genome to identify the miRNAs target positions (Table 1). Nucleotide sequences of the SCBGAV genome and sugarcane miRNAs were recorded in FASTA format to process further using computational algorithms with defined parameters. A detailed workflow pipeline was designed to identify miRNA binding sites within the SCBGAV genome (Fig. 2).

Table 1 miRNA target prediction algorithms.

Characteristic features considered by miRNA target prediction algorithms under study.

Algorithms	Main parameters	Availability	
miRanda	Seed pairing, target site accessibility, multiple target sites and translation inhibition	Web server and source code (http://cbio.mskcc.org/miRNA2003/miranda.html)	
RNA22	Seed pairing, target site accessibility and multiple target sites	Only Web server (https://cm.jefferson.edu/rna22/Interactive/)	
RNAhybrid	Seed pairing, target site accessibility, multiple target sites and translation inhibition	Web server and source code (http://bibiserv.techfak.uni-bielefeld.de/rnahybrid)	
psRNATarget	Target site accessibility, multiple target sites and translation inhibition	Only Web server (http://plantgrn.noble.org/psRNATarget/)	
Tapirhybrid	Seed pairing, target site accessibility and multiple target sites	Web server and source code (http://bioinformatics.psb.ugent.be/webtools/tapir/)	
Targetfinder	Seed pairing	Only source code	
Target-align	Multiple target sites	Web server and source code (Xie & Zhang, 2010)	

Figure 2 A schematic pipeline depicting the strategy of miRNA prediction from SCBGAV genome.

A schematic representation of miRNA target prediction in SCBGAV genome pipeline. Biological data set contains the kinds of data obtained for this study from NCBI (badnavirus genome) and miRBase (miRNAs). Computational algorithm group enlists three kinds of tools for the prediction of miRNA, secondary structure and miRNA-target interaction. R language was used to make plots and select data using in-house scripts/codes.

miRanda

miRanda is a computational algorithm (John et al., 2004) used to predict and identify the potential plant genomic miRNA targets (Archak & Nagaraju, 2007). It was designed to consider the following algorithm properties: sequence complementarity, RNA-RNA duplex dimerization, free energy and cross-species evolutionary conservation of molecular target sites. These properties yield output to weigh all matches and mismatch scores for gap penalties and for base pairs. It also promotes the prediction of multiple miRNA target sites including the ones with imperfect binding in the seed region within the 3′ UTR of the target site thereby enhancing the specificity (Betel et al., 2008; Witkos, Koscianska & Krzyzosiak, 2011). miRanda software was downloaded from the source website (http://cbio.mskcc.org/miRNA2003/miranda.html), and the SCBGAV genome was assessed to determine whether there were any possible targets for sugarcane miRNAs. The miRanda algorithm was run after defining standard settings. These include: Gap Open Penalty = −9.0, Score Threshold = 130, Minimum free energy (MFE) threshold (= −15.00 kcal/mol). MFE is an important statistical parameter to screen potential targets.

RNA22

RNA22 is a novel diverse web server (http://cm.jefferson.edu/rna22v1.0/), designed to implement a pattern-based approach to detect potential miRNA target sites (Miranda et al., 2006). The minimum folding energy (MFE) is a key parameter to screen possible miRNA without a cross-species conservation filter (Loher & Rigoutsos, 2012). Specificity and sensitivity values were selected at 61% and 63%, respectively. The value of the seed size was selected as 7 with unpaired base 1 permitted inside the seed region. There was no limit set to the maximum number of G:U wobbles in the seed region. The paired bases with minimum number and MFE were selected at 12 and −13.50 kcal/mol, respectively.

Tapirhybrid

Tapirhybrid is a novel web server (http://bioinformatics.psb.ugent.be/webtools/tapir) designed for the prediction of plant microRNA targets, including microRNA target mimics, using a very fast and precise algorithm (Bonnet et al., 2010). It considers many important parameters including seed pairing, multiple target sites and target site accessibility. It is one of the most highly recommended tools for plant microRNA prediction due to its great accuracy (Srivastava et al., 2014). It was designed to run using the RNAhybrid algorithm for microRNA target prediction and produces miRNA target scores (on the basis of the number of mismatches), the number of GU pairs, gaps and mismatches in the seed region. It is regarded as a high precision algorithm to enhance the accuracy of the predicted results. The setting for input parameters were standardized as score < =8 and MFE_ratio > =0.5.

RNAhybrid

RNAhybrid is a novel algorithm used to predict hybridization of miRNA and mRNA that is based on MFE (http://bibiserv.techfak.uni-bielefeld.de/rnahybrid). Site complementarity, target-site abundance and MFE are the unique parameters of RNAhybrid algorithm. MFE was set a threshold of −20 kcl/mol. RNAhybrid can also assign a p-value for the miRNA: miRNA interaction based on the number of binding sites within the 3′ UTR sequence, which is a measure of target-site abundance. The other filters remained fixed as default parameters (Krüger & Rehmsmeier, 2006).

psRNATarget

psRNATarget is a new web server (http://plantgrn.noble.org/psRNATarget/) that was developed to predict small RNA (sRNA) in plants. It was used to analyze complementary matching between target mRNA sequence and sRNA sequence based on a scoring schema (Dai, Zhuang & Zhao, 2018). Evaluation of target-site accessibility on the basis of UPE (unpaired energy) is another important feature of the psRNATarget algorithm. The standard parameter set for our analysis was as follows: penalty for (extending gap = 0.5, opening gap = 2, G.U pair = 1, other mismatches = 1), HSP size = 19, seed region = 2–7 nucleotides. The minimum expectation score was 7.0.

RNAfold algorithm

RNAfold algorithm was used to generate the stable secondary structure of pre-miRNAs from the precursors of the mature miRNAs (Lorenz et al., 2011). The parameters set for secondary structure prediction were: minimum free energy and partition function; avoid isolated base pairs and dangling energy on both sides of helix in any case; RNA parameters; rescale energy parameters at a given temperature 37C; interactive RNA secondary structure plot; RNA secondary structure plots with reliability annotation (partition function folding) and mountain plot (Turner & Mathews, 2009). Further, these predicted secondary structures were manually curated using rules reported by Zhang et al. (2005).

Circos algorithm

A Circos plot was developed between sugarcane-encoded miRNA and SCBGAV genes by applying the Circos algorithm (Krzywinski et al., 2009).

Statistical analysis

Predicted miRNA data obtained by applying four diverse algorithms were further analyzed using R-Language (v3.1.1, software version 3.5.1) (Gandrud, 2016). The graphical representation of the predicted miRNAs was processed using in house-scripts (readxl and ggplot2 packages).

Results

miRNA target prediction in genome of SCBGAV

In order to predict sugarcane miRNA targets in the SCBGAV genome, a combination of above mentioned computational algorithms were employed to magnify high accuracy of miRNA target prediction. This strategy also filters out the false-positive results. miRanda (John et al., 2004) was selected to implement various miRNA target prediction parameters. These are target site conservation and prediction of several miRNA target binding sites. Afterwards, RNA22 was widely used for pattern-recognition in the target sequence (Miranda et al., 2006). RNAhybrid is a unique algorithm which determines miRNA and mRNA hybridization using following unique parameters (Krüger & Rehmsmeier, 2006). This algorithm is applied to calculate (MFE) minimum free energy and mode of target inhibition suggested as per the Brodersen et al. (2008) conclusion. psRNATarget is the most widely used plant miRNA target prediction tool which deals with a large set of data. It was designed to predict plant sRNA targets by analyzing complementary matching between sRNA and target mRNA sequence and by evaluating target site accessibility (Dai, Zhuang & Zhao, 2018). No sugarcane miRNAs were predicted in the SCBGAV genome using Tapirhybrid algorithm. Figure 3 represents all the genomic positions targeted by sugarcane miRNAs by the various algorithms applied (Tables S2, S3 and File S1).

Figure 3 miRNA target prediction results of Sugarcane Bacilliform Guadeloupe A Virus.

(A) Target prediction results from miRanda, (B) target prediction by RNA22 (C) target prediction by RNAhybrid and (D) target prediction by psRNATarget.

Prediction of sugarcane-encoded miRNA targets in the SCBGAV ORF1

There is little information available about the function of this protein. ORF1 (538–1,071 nucleotides) encodes a hypothetical protein represented as P1 (177 amino acids). RNAhybrid predicted binding of sof-miR159e at locus 733, ssp-miR169 at locus 752, ssp-miR437 (a, c) at locus 978, and ssp-miR444 (a, b, c-3p) at locus 774 (Fig. 3C). In addition, psRNATarget predicted hybridization of ssp-miR166 at locus 981, ssp-miR444a at locus 775, and ssp-miR444b at position 756 (Fig. 3D).

Identification of miRNA targets in the SCBGAV ORF2

The DNA binding protein, represented as P2, is encoded by the ORF2. ORF2 has the least number of predicted targets by sugarcane miRNAs; only three miRNA of sugarcane (sof- miR167 (a, b) and ssp-miR528) were targeted at common locus 1,301, indicated by RNA22 (Fig. 3B). Similarly, RNAhybrid predicted bindings of sof-miR167 (a, b) at locus 1,304 and ssp-miR1432 at locus 1301 (Fig. 3C).

miRNA target prediction in the ORF3

The polyprotein is encoded by the ORF3 which contains several functional units. These proteins are coat, movement, aspartic protease, RT, and ribonuclease H (Bouhida, Lockhart & Olszewski, 1993; Geijskes et al., 2002; King et al., 2011). The RT/RNase H coding region is a commonly used taxonomic marker for badnaviruses. It is considered sufficient to address the sequence diversity present within Badnavirus genomes (Bousalem, Douzery & Seal, 2008). For the ORF3 of SCBGAV, seven different kinds of miRNAs were predicted by miRanda: (sof-miR159 (a, b, c, d, and e), sof-miR168a, sof-miR396, ssp-miR166, ssp-miR827, ssp-miR1128 and ssp-miR1432) (Fig. 3A). RNA22 predicted hybridization of the following miRNAs: sof-miR168 (a, b), sof-miR396, sof-miR408 (a, b, c and d) and ssp-miR444 (a, b) (Fig. 3B). Suitable miRNAs for targeting ORF3 were hypothesized by RNAhybrid to interact with the ORF3 of SCBGAV. These are sof-miR159 (a, b, d), sof-miR168a, sof-miR396, sof-miR408 (a, b, c, d and e), ssp-miR166, ssp-miR827, and ssp-miR1128 (Fig. 3C). Moreover, sof-miR159 (a, b, c, d), sof-miR167 (a, b), sof-miR396, ssp-miR528, ssp-miR444b and ssp-miR444c-3p were identified by psRNATarget (Fig. 3D).

Visualization of miRNA target interaction

A Circos plot was used to study the visualization of sugarcane miRNAs with SCBGAV-targets (ORFs) that display particular evidence for potential targets. We have for the first time reported sugarcane-encoded miRNAs and their targets simultaneously constructed in this manner (Fig. 4).

Figure 4 Circos plot showing network interaction between sugarcane-encoded miRNAs and their SCBGAV targets.

Sugarcane miRNAs (predicted by a consensus of algorithms) for SCBGAV silencing

Among all the predicted miRNAs for SCBGAV silencing, only four miRNAs (sof-miR167 (a, b), sof-miR396 and ssp-miR528) were predicted by all of the algorithms used (Fig. 5, Table 2 and Table S4). We have verified the efficacy of these miRNA targets against SCBGAV following suppression of RNAi-mediated badnavirus control through translational inhibition or cleavage of viral mRNA. Moreover, six miRNAs (sof-miR167 (a, b) at locus 5846 and 1310, sof-miR168a at locus 5506, sof-miR396 at locus 3394, ssp-miR444a at locus 775 and ssp-miR1128 at locus 6148) were predicted at the common locus by at least two of the algorithms used (Fig. 6). Out of 28 sugarcane miRNAs, only one miRNA (sof-miR396) was predicted by at least three algorithms used to have binding site at the same locus position 3394 (Table 3).

Figure 5 Venn diagram representing common sugarcane miRNAs predicted by all algorithms.

Prediction of secondary structures

Stable secondary structures of the consensus sugarcane miRNA precursor sequences was obtained using RNAfold algorithm. The precursor sequences were manually curated. Top seven stable secondary structures (based on MFEI value) of consensus sugarcane miRNAs are shown (Fig. 7). We have characterized the salient features of precursor miRNA, such as MFE (minimum free energy) to determine the secondary structure of nucleic acids. In our studies, MFE range from −56.1 to −101.7 kcal/mol, GC content (42–65%) and MFEI ranges from −0.7 to −1.17 (Table 4).

Table 2 miRNA-target pairs were selected from miRanda analysis.

Locus position was selected by consensus analysis of at least two algorithms.

SugarcanemicroRNAs	miRNA-target pair	Locusposition	MFE*(kcal/mol)	
sof-miR167(a, b)	Query: 3′ gucUAGUACGACCGUCGAAGu 5′

Ref: 5′ aaaATCAAGTT-GCAGCCTCa 3′	5846–5865	−26.60	
sof-mi396	Query: 3′ guCAAGUUC-UUUCGACACCUu 5′

Ref: 5′ agGATTAGGTGATGCTGTGGAg 3′	3394–3415	−24.90	
ssp-miR528	Query: 3′ gaggAGACGUACGGGGAAGGu 5′

Ref: 5′ gcgaTCCGC-CCCCCCTTCC- 3′	7426–7444	−28.0	
Notes.

MFE (Minimum Free Energy)* and mode of target Inhibition** was determined by RNAhybrid.

Discussion

For possible miRNA target prediction in the genome of SCBGAV, a combination of the aforementioned computational tools was used in order to filter out the false-positive results and to increase the accuracy of miRNA target prediction. miRanda was implemented to validate various parameters, from target site conservation to whole genome prediction of miRNA target genes (John et al., 2004; Riffo-Campos, Riquelme & Brebi-Mieville, 2016). Then, RNAhybrid and psRNATarget were used, both of which are strongly prescribed for plant miRNA target identification (Dai, Zhuang & Zhao, 2018; Srivastava et al., 2014; Zhang & Verbeek, 2010). RNA22 is a novel algorithm that applies an in-silico strategy that is highly divergent in comparison to other algorithms. Pattern-based recognition is the key feature of this algorithm (Min & Yoon, 2010; Miranda et al., 2006).

Figure 6 Intersection plot showing sugarcane miRNAs predicted from at least two algorithms.

The results from this study suggest that SCBGAV is susceptible to targeting by consensus sugarcane-encoded miRNAs. The genome components of SCBGAV (ORF1, ORF2 and ORF3) seemed to be principally prone sites for sugarcane-miRNA regulation. While in vivo demonstration requires validating functional efficacy, the degree of complementarity between target mRNA and miRNA concludes the fate of the predicted sites. A full complementarity binding between target mRNA and miRNA sequence results in endonucleolytic cleavage and disruption. Contrary to this, partial target-site complementarity characteristically down-regulates target-gene expression by suppressing translation of target mRNA.

The present study identifies suitable sugarcane-encoded miRNAs to exhibit a stronger degree of target-site complementarities within the ORF1, ORF2 and ORF3 of SCBGAV. These predicted miRNAs may be utilized to develop effective amiRNA constructs, which could be used to enhance the sugarcane immunity to SCBGAV. Pairing multiple miRNAs with a single mRNA induces effective RNA silencing (Doench & Sharp, 2004). In the current study, we have predicted several miRNA targets which were associated with SCBGAV (ORF1, ORF2, and ORF3) genes at multiple loci. A deeper understanding of these vital ORFs involved in SCBGAV epidemic via miRNA-mediated control of gene expression would significantly assist in the development of molecular approaches to combat the dissemination of SCBGAV. The miRNA-target pairs ensuring MFE exceeding the threshold standards were predicted.

Table 3 Sugarcane miRNAs and their target positions in the SCBGAV genome identified by all four algorithms.

miRNA name	Position miRanda	Position RNA22	Position RNAhybrid	Position psRNATarget	MFE* miRanda	MFE** RNA22	MFE RNAhybrid	Expectation psRNATarget	
sof-miR156			7104				−23.2		
sof-miR159a	5282		1659	3779	−17.18		−25.4	6	
sof-miR159a(1)	6739				−18.11				
sof-miR159b	5282		1659	3779	−17.18		−25.4	6	
sof-miR159b(1)	6739				−18.11				
sof-miR159c	6739		6896	3779	−16.70		-28	6	
sof-miR159d	5282		1659	3779	−17.18		−25.4	6	
sof-miR159d(1)	6739				−18.11				
sof-miR159e	5282		733		−15.81		−25.5		
sof-miR167a	5846	1310	1304	5846	−16.08	−15.80	−26.6	7	
sof-miR167b	5846	1310	1304	5846	−16.08	−15.80	−26.6	7	
sof-miR168a	5506	5506	6084		−21.24	−17.70	−25.6		
sof-miR168a(1)	7050	7050			−18.45	−17.10			
sof-miR168a(2)		5137				−14.30			
sof-miR168b	7050	7050	6937		−20.05	−19.40	−25.6		
sof-miR168b(1)		5506				−13.70			
sof-miR396	3394	3394	3394	5732	−19.99	−17.80	−24.9	5.5	
sof-miR396(1)		4115				−14.20			
sof-miR408a		1735	5136			−13.70	−27.6		
sof-miR408b		1735	5136			−13.70	−27.6		
sof-miR408c		1735	5136			−13.70	−27.6		
sof-miR408d		1735	5136			−13.70	−27.6		
sof-miR408e			6152				−28.9		
ssp-miR166	3249		5572	981	−21.45		−26.7	6.5	
ssp-miR166(1)	5589				−17.54				
ssp-miR166(2)	5761				−21.74				
ssp-miR169			752				−24.8		
ssp-miR437a			978				−21.7		
ssp-miR437b			6829				−21.8		
ssp-miR437c			978				−24.3		
ssp-miR528	7426	1310	7407	2897	−18.37	−13.70	-28	6.5	
ssp-miR528(1)	7395				−18.05				
ssp-miR827	5378		3010		−16.40		-24		
ssp-miR444a		4103	774	775		−16.30	−27.6	6.75	
ssp-miR444b		4103	774	756		−16.30	−27.6	6.0	
ssp-miR444b(1)				2615				6.5	
ssp-miR444b(2)				5009				6.5	
ssp-miR444b(3)				5731				6.5	
ssp-miR444c-3p			774	4940			−26.7	6.5	
ssp-miR1128	6141		6148		−22.96		−27.8		
ssp-miR1432	6070		1301		−15.48		−20.4		
Notes.

*MFE: Minimum free energy measured in /Kcal/mol where *MFE represents minimum folding energy measured in Kcal/mol.

Our study is unique from previous studies in that it provides an in silico approach for comprehensive analysis of host-derived miRNAs using seven computational algorithms and advanced statistical approaches. Previous studies that have identified plant-encoded miRNA Targets in plant viruses have been limited by small numbers of miRNAs screened, small sample sizes, lack of independent validation sets, and the use of inappropriate statistical methods to present miRNA interaction data. Notably, we have validated the efficacy of the consensus seven sugarcane-encoded miRNA targets (detected two algorithms at the same locus) against SCBGAV following suppression of RNAi-mediated viral combat. These are achieved by translational inhibition or cleavage of viral miRNA.

Figure 7 Stable secondary structures of top seven pre-miRNAs, precursors of the mature sugarcane miRNAs found in the study as the miRNAs detected by a consensus of algorithms.

Names with minimum free energy (MFE) are represented: (A) sof-MIR396 (−67.40 Kcal/mol) (B) sof-MIR167a (−82.70 Kcal/mol) (C) sof-MIR167b (−86.00 Kcal/mol) (D) sof-MIR168a (−66.20 Kcal/mol), (E) sof-MIR168b (−56.10 Kcal/mol), (F) ssp-MIR444a (−57.70 Kcal/mol), (G) ssp-MIR1128 (−101.70).

Table 4 Predicted characterized features of secondary structures of precursor miRNAs.

miRNA ID	miRNA sequence	Length miRNA	Length precursor	MFE1 (Kcal/mol)	AMFE2	MFEI3	(G + C)%	
sof-miR396	UUCCACAGCUUUCUUGAACUG	21	134	−67.40	−50.29	−1.17	42.85	
sof-miR167a	UGAAGCUGCCAGCAUGAUCUG	21	188	−82.70	−43.98	−0.83	52.38	
sof-miR167b	UGAAGCUGCCAGCAUGAUCUG	21	188	−86.00	−45.74	−0.87	52.38	
sof-miR168a	UCGCUUGGUGCAGAUCGGGAC	21	104	−66.20	−63.65	−1.02	61.90	
sof-miR168b	UCGCUUGGGCAGAUCGGGAC	20	103	−56.10	−54.46	−0.83	65.00	
ssp-miR444a	UGCAGUUGUUGCCUCAAGCUU	21	105	−57.70	−54.95	−1.14	47.61	
ssp-miR1128	UACUACUCCCUCCGUCCCAAA	21	275	−101.70	−36.98	−0.70	52.38	
Notes.

1MFE represents minimum free energy. 2AMFE is an adjusted minimum free energy calculated by MFE/length of precursor*100. MFEI defines as minimum free energy index which was calculated by AMFE/(G + C) %.

Applying a combination of four diverse computational algorithms (miRanda, RNA22, Tapirhybrid and psRNATarget), the predicted consensus results are highly reliable and robust for assessing interaction between sugarcane miRNAs and viral genome (Oliveira et al., 2017). This miRNA-viral gene interaction was further verified by RNAhybrid (standard MFE predicted) and Circos algorithms (network visualization). False-positive miRNA-target interactions predicted by computational algorithms depend upon the mechanism of the miRNA-target recognition (Pinzón et al., 2017). In addition, MFE is one of the key factors to affect the miRNA-targets interactions for results sorting. A lower MFE value has high potential to develop miRNA-target complex (Hammell et al., 2008; Kertesz et al., 2007). We have applied a stringent cut-off −15 kcal/mol for miRanda and −20 kcal/mol for RNAhybrid to narrow down miRNA candidates.

In order to control false-positive prediction, union and intersection approach was applied. Union approach based on grouping of one or more algorithmic tools which results increase of number of true targets as well the number of false targets. It decreases specificity at the cost of increasing sensitivity. Using intersection approach relies on the mixing two or more tools which improves the specificity at the cost of decreasing sensitivity (Witkos, Koscianska & Krzyzosiak, 2011). Our analysis indicated that both the strategies showed the highest performance as shown in (Figs. 5 and 6). These miRNAs were identified and considered as the most desirable sugarcane miRNAs against the genome of SCBGAV. These predicted miRNAs were identified after setting the parameters of minimum free energy, seed pairing, target site accessibility, folding energy and pattern recognition, thus integrating all aspects of miRNA target prediction. Therefore, they are the most suitable selections for gene silencing.

Similar studies have been published previously using multiple computational algorithms to identify a best fit miRNA for a desirable target to understand plant-virus interaction. First time, genome-wide identification and comprehensive analysis was carried out to predict maize (Zea-mays)-encoded miRNA targets against maize chlorotic mottle virus (MCMV) (Iqbal et al., 2017). We have adopted the same novel computational strategy to predict the sugarcane-encode miRNA targets in the SCBGAV genome to combat badnaviruses in sugarcane.

As the amiRNAs have high specificity to the designed target gene, detrimental off-target effects can be minimized, permitting their silencing expression to be stably transmitted to future generations (Ossowski, Schwab & Weigel, 2008; Zhao et al., 2009). Furthermore, the small size of amiRNA permits for the insertion of multiple and distinct amiRNAs within a single gene expression cassette, which can then be transformed to develop transgenic plant resistant to multiple viruses simultaneously (Niu et al., 2006; Park, Zhai & Lee, 2009; Schwab et al., 2010). We have designed future work to validate this promising amiRNA-based strategy can in fact be used to develop durable SCBGAV- resistance in transgenic sugarcane.

Conclusions

The current study details an organized computational approach for the identification of host-derived miRNAs aimed at silencing viral genes affecting the sugarcane host plant by amiRNA-mediated interference targeting different genes of the SCBGAV. This study offers an enhanced means to computationally investigate the best-candidate miRNAs against badnaviruses, prior to cloning. As our approach allows a narrow-range of match-mismatch in microRNA-mRNA attachment, it screens most of the falsely predicted attachments. These predicted miRNAs were identified after setting the parameters of MFE, seed pairing, target site accessibility, folding energy and pattern recognition, thus using key features of miRNA target prediction. Most importantly, we identified six miRNAs (sof-miR167 (a, b), sof-miR168a, sof-miR396, ssp-miR444a and ssp-miR1128) that could target the SCBGAV genome. Among them, sof-miR396 was predicted as the top-ranking effective candidate, capable of targeting the vital ORF3. To further refine our results, we also identified key interaction of miRNA networks and target genes by constructing Circos graph of consensus miRNAs and their targets. This short-listed miRNA is the best candidates to be utilized in sugarcane plant transformation for the development of SCBGAV-resistant sugarcane cultivars. These results offer an evidence of idea for the construction of innovative amiRNA-based therapeutics against emerging SCBGAV badnaviruses.

Supplemental Information

File S1 Computational prediction of Sugarcane-encoded MicroRNAs in the Genome of SCBGAV using all algorithms

Click here for additional data file.

Table S1 Characteristic features of sugarcane miRNAs retrieved from miRBase database

Click here for additional data file.

Table S2 Prediction of sugarcane miRNAs in the SCBGAV genome

Sugarcane-encoded miRNAs predicted from all four algorithms. Genome positions and the output parameters including minimum free energy, folding energy and expectation have been included.

Click here for additional data file.

Table S3 Sugarcane-encoded miRNA target binding sites locus position in each gene of SCBGAV

Click here for additional data file.

Table S4 Identification of common sugarcane miRNAs in the SCBGAV genome

Consensus sugarcane-encoded miRNA were selected from all four algorithms.

Click here for additional data file.

We are highly thankful to our lab colleagues for their assistance in data analysis. We thank to Dr. Zhiqiang Xia (ITBB) for assistance in constructing Circos plot. We are thankful to our senior lab colleagues Xiao-Yan Feng and Lin-Bo Shen for reading the revised manuscript as well as the editor and two referees for their valuable comments on this manuscript. We are highly thankful to a qualified native English speaking editor at the PeerJ Experts for editing proper English language.

Additional Information and Declarations

Competing Interests

Author Contributions

Data Availability

The authors declare there are no competing interests.

Fakiha Ashraf and Xiaowen Hu analyzed the data, performed the experiments, prepared figures and/or tables, authored or reviewed drafts of the paper, and approved the final draft.

Muhammad Aleem Ashraf analyzed the data, conceived and designed the experiments, performed the experiments, prepared figures and/or tables, authored or reviewed drafts of the paper, and approved the final draft.

Shuzhen Zhang conceived and designed the experiments, authored or reviewed drafts of the paper, and approved the final draft.

The following information was supplied regarding data availability:

Saccharum officinarum L. miRNA data is available at miRBase: MI0001754–MI0001769, which can be accessed by species name.

Additional data is available at NCBI: NC_038382.1.

Tools used to analyze the data are available at the Jefferson Computational Medicine Center RNA22 (https://cm.jefferson.edu/rna22/), BiBiServ (https://bibiserv.cebitec.uni-bielefeld.de/rnahybrid), and psRNATarget (http://plantgrn.noble.org/psRNATarget/).

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
