# Peer review of "A novel computational approach to the silencing of Sugarcane Bacilliform Guadeloupe A Virus determines potential host-derived MicroRNAs in sugarcane (Saccharum officinarum L.)"

_PeerJ, doi:10.7717/peerj.8359_

## Round 0.1 · original submission · Major Revisions

The manuscript got two reviews containing some serious remarks on the paper novelty and result section. I believe it is original work presenting original method. However some concern is about exiting method published previously in Frontiers - https://www.frontiersin.org/articles/10.3389/fpls.2017.00372/full

I think here is novel application (sugarcane vs maize), however please describe and accurately cite the existing methods.

Reviewer 1 ·

Basic reporting

The authors have characterized the S.officinarum miRNAs that potentially target SBCGAV virus. They reported miR396 as a candidate miRNA to base the generation of amiRNA to combat SBCGAV. The background of the manuscript is poor and showed complete unfamiliarity of miRNAs features. The figures were also very poor and lacking information. The whole manuscript needs a great language improvement. The analysis performed do not support the results and conclusion. The authors based their research in a computational prediction of candidate miRNAs targets. However, the pairing between miRNAs:SBCGAV ORF was very bad. There is no evidence in the literature of plant canonical miRNA targeting of virus ORFs. Besides, they propose that S.officinarum resistance to SBCGAV is determined by miRNAs expression. While resistance is a very complex feature that are based in many different genes. The supplementary data was not cited in the manuscript. The authors must follow the structure of the journal PeerJ. The manuscript is lacking in many aspects and in this format is not suitable for publication.

Experimental design

The research question was weak. The applied methods were not robust to the study proposal.

Validity of the findings

The data was inconclusive. The conclusions presented are very questionable, since they do not showed a correlation between S.officinarum resistance to SCBGAV virus and expression of endogenous miRNAs. The study is very weak on the proposal of application of amiRNA to combat SCBGAV virus.

·

Basic reporting

Language is ok only minor errors of not using articles a many places "a" and "the", not considering the plurality of the matter under discussion and a few others
e.g.and not limited to
line 102 permitted inside "the" seed region, line 108 ->"a threshold"
line 123 was ->were
line 138 binding -> bindings
false positive -> false-positive

similar studies have been published using multiple tools to find a best fit miRNA for a target in plant and virus interaction, authors should discuss how this study have a novel approach and how it differs from the other studies published on the same pattern in its novelty.

Tables are less organized (acceptable with few a changes) to show the characteristics of a single locus in the same row of an excel entry. it will be much easier for readers to verify the claims which have been made in this study. Figure quality should be increase greatly. Text shown in figures are almost unreadable specially fig 1 and 2. However, figures are showing correct positions of target sites and other energy related parameters.

Experimental design

Authors aimed to predict microRNA/s which could be used for a resistant GMO against this virus. Overall approach is very good that the authors tried to cover all the aspects of attachment parameters with its target computationally.
Objective is well defined.

Validity of the findings

Authors have concluded the study that 3 out of four tools used have predicted that miR396 is the best candidate miRNA to be considered for making a GMO against this virus. Data provided is correct and all tools have predicted the potential for this possible attachment.
However, in conclusion section author have made a claim to have shortlisted more than one miRNAs which is not correct.
Authors should amend the conclusion section with respect to the findings of the study.

Additional comments

1. work using the same strategy has been published so it should not be titled as "novel approach".
2. graphs should be included in main file for an easy understanding and it will help the readers.

---

## Round 0.2 · Minor Revisions

The reviewers had no direct comments on the manuscript. But it still needs a revision.

To save time I recommend you check again the previous comments and reply not formally, but by making your work more readable by students, non-specialist, for general biologists. This is not a comment about the quality of the English, it is more about the readability so that you can reach the broadest audience.

This work has to be restructured for more clear presentation of the results. As the editor I summarize the remarks here in plain text.
Overall the research topic is of interest for PeerJ readers. But the presentation could be improved, especially in the figures.

Abstract text has to be rewritten to make it more interesting for wide readers’ audience.

Descriptive numbers should be changed to the results -

So, for example “A total of 28 potential mature target miRNAs were retrieved”
“Only 4 sugarcane miRNAs are selected...” are technical details
Just describe methods and main results.

Some phrases about agricultural importance of this pathogen of sugarcane would improve abstract presentation.

Avoid passive voice in the phrases -
“A Circos plot was created..”
Show first the results.
It is better, for example write “interaction network of sugarcane-encoded miRNAs with SCBGAV genes, created using Circos, allow analyze new targets.”..

Conclusion is not clear -
“The present study concludes a comprehensive report...”
Why it is comprehensive?

In the Introduction please write more about sugarcane bacilliform virus research history, practical application in agriculture.. I recommend extend this part.

In the Materials&Methods section describe parameters of miRNA extraction from miRBase. Why only 28? If change parameters?
Methods (software) are described in details. But why “Four computational algorithms ... named miRanda, RNA22, RNAhybrid and psRNATarget”? Are they complementary to each other? Why not use, say, only two algorithms?

What are recommendations for other researchers for other model organism to search miRNA targets? Is here any specificity for sugarcane virus?

Conclusion section could be extended - “This short-listed miRNA is the best candidates to be utilized..” It means the work is not finished. What is the practical outcome for plant biology research?

The Figures has to be improved.

Figure 1 is of low resolution. Please increase resolution of the graphics.

Figure 2 - Pipeline is unidirectional. But all the software are given together in the scheme. Circos should be shown separately, at least.

Figure 3 - Axes X and Y have legends in small font size, not visible. The dots on the plots could be given larger. Not clear what one can see in the plot. Just prediction at different genome positions?
Remove grey background in the figure.
The same is for Figure 6

Figure 5 (Circos plot) - increase resolution.

Figure 7 - resolution.
Need some comments here about RNA secondary structure energy.

Table 1. Column “Availability” - add web-links there. Otherwise the table has no sense.

Table 2. “Inhibition” column is redundant.
I suggest remove it from the table. Just mention Cleavage inhibition it in the text.

Please fix these presentation details of the manuscript corresponding to the publication standards. I believe it could be accepted at next reviewing round.

---

## Round 0.3 · accepted · Accept

We have no more critical comments from the reviewers. As I see in the updated files all the comments were taken into account. As academic editor I endorse publication of this work in the current form.